# HiVG: Hierarchical Multimodal Fine-grained Modulation for Visual Grounding

### Linhui Xiao
[1]MAIS, Institute of Automation,
Chinese Academy of Sciences
[2]Pengcheng Laboratory
[3]School of Artificial Intelligence,
University of Chinese Academy of
Sciences
xiaolinhui16@mails.ucas.ac.cn

### Xiaoshan Yang
[1]MAIS, Institute of Automation,
Chinese Academy of Sciences
[2]Pengcheng Laboratory
[3]School of Artificial Intelligence,
University of Chinese Academy of
Sciences
xiaoshan.yang@nlpr.ia.ac.cn

### Fang Peng
[1]MAIS, Institute of Automation,
Chinese Academy of Sciences
[2]Pengcheng Laboratory
[3]School of Artificial Intelligence,
University of Chinese Academy of
Sciences
pengfang21@mails.ucas.ac.cn

### Yaowei Wang
[1]Pengcheng Laboratory
[2]Harbin Institute of Technology
(Shenzhen)
wangyw@pcl.ac.cn

### Changsheng Xu*
[1]MAIS, Institute of Automation,
Chinese Academy of Sciences
[2]Pengcheng Laboratory
[3]School of Artificial Intelligence,
University of Chinese Academy of
Sciences
csxu@nlpr.ia.ac.cn

## ABSTRACT

Visual grounding, which aims to ground a visual region via natural language, is a task that heavily relies on cross-modal alignment. Existing works utilized uni-modal pre-trained models to transfer visual or linguistic knowledge separately while ignoring the multimodal corresponding information. Motivated by recent advancements in contrastive language-image pre-training and low-rank adaptation (LoRA) methods, we aim to solve the grounding task based on multimodal pre-training. However, there exists significant task gaps between pre-training and grounding. Therefore, to address these gaps, we propose a concise and efficient hierarchical multimodal fine-grained modulation framework, namely HiVG. Specifically, HiVG consists of a multi-layer adaptive cross-modal bridge and a hierarchical multimodal low-rank adaptation (HiLoRA) paradigm. The cross-modal bridge can address the inconsistency between visual features and those required for grounding, and establish a connection between multi-level visual and text features. HiLoRA prevents the accumulation of perceptual errors by adapting the cross-modal features from shallow to deep layers in a hierarchical manner. Experimental results on five datasets demonstrate the effectiveness of our approach and showcase the significant grounding capabilities as well as promising energy efficiency advantages. The project page: https://github.com/linhuixiao/HiVG.

---

*Corresponding author.

## CCS CONCEPTS

• **Computing methodologies** → **Computer vision tasks**; **Scene understanding**.

## KEYWORDS

Multimodality; Visual Grounding; Referring Expression Comprehension; Low-Rank Adaptation; Hierarchical

**ACM Reference Format:**
Linhui Xiao, Xiaoshan Yang, Fang Peng, Yaowei Wang, and Changsheng Xu. 2024. HiVG: Hierarchical Multimodal Fine-grained Modulation for Visual Grounding. In *Proceedings of the 32nd ACM International Conference on Multimedia (MM '24), October 28-November 1, 2024, Melbourne, VIC, Australia.* ACM, New York, NY, USA, 10 pages. https://doi.org/10.1145/3664647.3681071

## 1 INTRODUCTION

Visual Grounding (VG), also known as Referring Expression Comprehension (REC) or Phrase Grounding (PG) [9, 19, 29, 41, 47, 62, 63, 75], is a fundamental and challenging task at the intersection fields of vision-language understanding, which can be potentially used in a wide range of applications [1, 6, 35], such as visual question answering [1], human-machine interaction [6] *etc.*. Unlike object detection [37, 38], which requires a predefined and fixed set of categories, grounding is not limited to specific categories but instead needs to identify the specific image region according to the language expression semantics. Thus, grounding is a task that strongly relies on the interaction and alignment of multimodal features.

Existing state-of-the-art (SOTA) approaches [9, 10, 16, 55, 71, 76, 78] utilize uni-modal pre-trained detection models or language models (*e.g.*, ResNet [15], Swin Transformer [39], DETR [4], ViT-Det [28], BERT [11], RoBERTa [36] *etc.*) to facilitate grounding learning. These methods separately transfer the language or vision knowledge from pre-trained models by using resource-consuming fully parameter fine-tuning, ignoring the multimodal corresponding

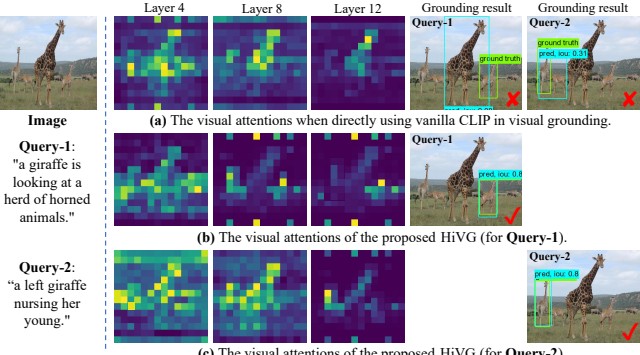

**Figure 1: Visual attentions and grounding results of CLIP and the proposed HiVG. The attentions are perceived by the [CLS] token over vision tokens.**

information. Therefore, it is natural for us to consider using cross-modal pre-trained models as a solution to the grounding problem.

By utilizing language supervision from large-scale unlabeled data, Vision-Language Pre-training (VLP) can acquire comprehensive multimodal representations. Recently, the remarkable success of Contrastive Language-Image Pre-training (CLIP) [48] has demonstrated its ability to learn general visual concepts, which assists many multimodal tasks to achieve remarkable improvements [23, 44, 48, 61]. In visual grounding, there are also works, *e.g.*, CLIP-VG [63] and Dynamic-MDETR [53], which consider using CLIP. However, existing methods mainly utilize the CLIP as a backbone to extract strong vision and language features, without comprehensively investigating on the significant task gap between the pre-trained CLIP and the downstream grounding, which hinders exploiting the full potential of pre-training models. In this work, we scrutinize the task gap from two aspects. **(1) *Data bias.*** There inevitably exists a certain bias in data between the large-scale pre-training and grounding. Directly utilizing the frozen vision backbone of the CLIP may extract visual features sensitive to general objects that are not the focus of the query in visual grounding. For example, as shown in Fig. 1-(a), the middle giraffe receives highlight attention, but it has little relation to the grounding task. **(2) *Difference in learning objectives.*** The visual grounding task needs to find the precise image region that has the target object expressed by the query sentence. In contrast, CLIP works as a multimodal pre-trained model, which is only constrained to coarsely align noisy image and text data [51] in a self-supervised way. In addition, the self-supervised constraint is only performed at the final layer. When directly using the pre-trained CLIP in visual grounding, some valuable fine-grained visual information in the bottom vision layers may be discarded, which brings challenges for accurately locating the object box. For example, as shown in Fig. 1-(a), the left giraffe receives relatively small attention areas, which leads to inaccurate box of the target object.

It is not trivial to address the two kinds of task gaps. **(1) For the task gap of data bias,** extracting the features of the query text to guide the visual feature learning is a potential way to solve it. However, the query text has a feature space that is very different from the visual space and it is difficult to find the appropriate semantic information from the query features to guide the learning of different vision layers. **(2) For the task gap of learning objectives,** to

adapt the pre-trained CLIP to the grounding task, a straightforward way is fine-tuning the pre-trained weights. Whereas, this scheme may lead to catastrophic forgetting, which is harmful to retain the general knowledge learned by the pre-trained models. Another potential solution is to employ Low-Rank Adaptation (LoRA) [18] by fine-tuning only a few parameters. However, simply applying LoRA does not achieve fine-grained adaptation and even lead to performance degradation. Since high-level features depend on low-level features, and they are susceptible to perturbations of the shallow features. If all layers of a large-scale pre-trained model are adapted simultaneously, perceptual errors in bottom layers may accumulate and amplify. Therefore, it is necessary to consider a hierarchical approach for progressively adapt fine-grained visual features from shallow to deep layers.

In this paper, we propose a hierarchical multimodal fine-grained modulation framework to more effectively adapt the pre-trained CLIP to grounding, namely **HiVG**. It is a concise and efficient end-to-end framework that can alleviate two kinds of task gaps (*i.e.*, data bias and learning objectives) through a multi-layer adaptive cross-modal bridge and a hierarchical low-rank adaptation paradigm.

**Firstly**, to address the inconsistency between visual features of the pre-trained CLIP and those required for grounding, as well as establish a connection between multi-level visual and text features, we have designed a multi-layer adaptive cross-modal bridge. Specifically, the cross-modal bridge includes a sample-agnostic semantic weighting module and a multi-head cross-attention module. The weighting module incorporates learnable multi-level sample-agnostic adaptive weights, facilitating the selection of appropriate linguistic features through a residual operation. The multi-head cross-attention utilizes the selected multi-level text features for guiding the learning of the visual features required in grounding. The sample-agnostic semantic weighting scheme is inspired by [3, 8], *i.e.*, specific layers of a pre-trained model may have distinct responses to certain concepts or semantics that are independent of the input and relevant to the network layers.

**Secondly**, to prevent the accumulation of errors layer by layer in the downstream adaptation process of the pre-trained model, we propose Hierarchical Low-rank Adaptation (**HiLoRA**) paradigm. Existing methods mainly utilize LoRA[18] as a parameter-efficient fine-tuning (PEFT) method to learn a single round along with the entire model. Different from previous methods [18, 54], we divide the network layers of the pre-trained CLIP into multiple layer groups. The low-rank adaptation is allocated into multiple stages where each stage relates to several layer groups. Then, during the adaptation process, visual features are recursively and hierarchically adapted from shallow to deep layers in a hierarchical manner. Simultaneously, with the assist of the multi-layer cross-modal bridge, HiLoRA can not only achieve fine-grained hierarchical adaptation, but also enable the low-rank matrix perception based on the vision and language cross-modal information.

As show in Fig. 1-(b) and (c), benefiting from the hierarchical multimodal fine-grained modulation structure, HiVG exhibits heightened sensitivity towards visual region information, demonstrates enhanced comprehension of complex text, and significantly bridges the gap between pre-training and grounding tasks. Our method achieves SOTA performance on five widely used datasets, including RefCOCO/+/g [41, 75], ReferitGame [21] and Flickr30K

Entities [46]. HiVG outperforms the CLIP-based SOTA method, Dynamic-MDETR [53], on RefCOCO/+/g datasets by 3.15%(testB), 2.11%(testA), 4.30%(test), and also outperforms the strong detector-based SOTA method, TransVG++ [10], on the three datasets by 2.30%(testB), 3.36%(testA), 2.49%(test), respectively. Meanwhile, our model can obtain SOTA results on 224×224 small-resolution images without relying on high-resolution images (*e.g.*, 640×640) like other works [10, 55, 71]. Additionally, it significantly accelerates inference processes and is **8.2×** faster than TransVG++ (Fig. 4).

The main contributions can be summarized as three-fold:

- We proposed a concise hierarchical multimodal modulation framework, which utilizes the hierarchical structure to gradually adapt CLIP to grounding. HiVG achieves fine-grained interaction between multi-level visual representations and language semantics, and significantly alleviates the task gap between CLIP and grounding.
- We are the first to propose the hierarchical multimodal low-rank adaptation structure. HiLoRA is a basic and concise hierarchical adaptation paradigm, which is task-agnostic.
- We conducted extensive experiments to verify the effectiveness of HiVG approaches. Results show that our method achieves promising results, surpassing the SOTA methods under the same setting by a significant margin. Besides, our model offers significant computing efficiency advantages.

## 2 RELATED WORK

### 2.1 Visual Grounding

Visual grounding has recently received significant research attention, and it can be categorized into several settings. On the one hand, represented by TransVG [9], this setting involves full-parameter fine-tuning utilizing pre-trained closed-set detectors and language models. It is considered the most conventional and extensively studied setting. Under this setting, numerous complex two-stage [17, 31, 34, 74] and one-stage [68, 70, 77] methods emerged based on traditional detection networks in the early CNN era. After the introduction of ViT [12, 58], the Transformer-based networks [9, 10, 16, 20, 27, 40, 42, 55, 71, 72] constantly pushes the accuracy to new limits. However, these works only focus on achieving grounding by using independently pre-trained uni-modal detectors and language encoders while ignoring the alignment of cross-modality information within pre-trained model itself. More recent works, such as QRNet [71], VG-LAW [55], TransVG++ [10], *etc.*, only incorporate language-guided knowledge in vision backbone without attempting multi-level fine-grained alignment of multimodal features. Motivated by this setting, several works, such as CLIP-VG [63] and Dynamic-MDETR [53], have recently sprung up to the setting of fine-tuning with vision and language (VL) self-supervised pre-trained models. Following this setting, our study delves into a deeper perspective of hierarchical multimodal information and achieves fine-grained interaction of cross-modal features. On the other hand, with the evolution of the pre-training paradigm, many new settings have recently emerged that significantly improve the grounding performance, such as fine-tuning with box-level dataset-mixed open-set detection pre-trained models (*e.g.*, MDETR [20], Grounding-DINO [33], *etc.*), fine-tuning with box-level / multi-task mixup-supervised pre-trained models (*e.g.*, UniTAB [69], UNITER

[7], OFA [60], *etc.*), and grounding multimodal large language models (GMLLMs, *e.g.*, Shikra [6], Kosmos-2 [45], Ferret [73], LION [5], *etc.*). However, these works require a large amount of fine-grained labeled data, resulting in a relatively high training cost.

### 2.2 Contrastive Language-Image Pre-training

With the promotion of learning general and transferable cross-modal representations [14, 64–67], VLP has become the core training paradigm of modern VL research. Benefiting from self-supervised contrastive learning, CLIP has demonstrated impressive generalization and downstream transfer ability in a series of studies [44, 48]. More recently, some works utilized CLIP to realize grounding transfer, such as adapting-CLIP [26], ReCLIP [56] *etc.*, but these works are limited to using CLIP features as aids in an unsupervised or zero-shot setting [22, 56] and cannot directly perform grounding. Although CLIP-VG [63], Dynamic-MDETR [53], JMR [79] *etc.*, realizes grounding transfer, it does not conduct more in-depth research on the task gaps and the hierarchical cross-modal features. Unlike previous work, our study fills the gap by conducting a more comprehensive study of the cross-modal task gaps between CLIP's pre-training and downstream grounding.

### 2.3 Low-Rank Adaptation

LoRA [18] freezes the weights of pre-trained model and injects trainable rank decomposition matrices into each layer of the Transformer [59], thereby significantly reducing the number of trainable parameters for downstream tasks. Vanilla LoRA has been proposed in the field of natural language processing for Large Language Models (LLM) such as LLaMA2 [57], GPT-2 [49], GPT-3 [2] with 175B parameters, *etc.*. Recently, researchers have attempted to apply vanilla LoRA in the fields of cross-modal tasks [54]. However, since cross-modal tasks primarily emphasize the interaction of multi-modal information in contrast to unimodal language or visual tasks, the application of LoRA to grounding tasks remains unexplored. Consequently, we propose HiLoRA as an effective solution for addressing the existing gaps in multimodal downstream transfer.

## 3 METHODOLOGY

In this section, we propose our hierarchical multimodal fine-grained modulation framework for visual grounding, namely **HiVG**, which mainly consists of the multi-layer adaptive cross-modal bridge and the hierarchical low-rank adaptation (HiLoRA) paradigm. We will introduce each of these methods in the following sections.

### 3.1 Framework Overview

Our aim is to achieve fine-grained hierarchical cross-modal feature modulation, so as to narrow the task gap between the self-supervised pre-training and grounding. Therefore, we integrate the multi-level image and text representations from a hierarchical perspective with the facilitation of multi-layer adaptive cross-modal bridge and the hierarchical LoRA paradigm. Specifically, as shown in Fig. 2, the network architecture of HiVG consists of a CLIP image encoder, a CLIP text encoder, a grounding encoder and a regression head. Firstly, for any given image $\mathcal{I} \in \mathbb{R}^{3 \times H \times W}$ and text $\mathcal{T} \in \mathbb{R}^{L_l}$ pairs, the visual and text encoders encode the image and text tokens to obtain the visual feature $f_v \in \mathbb{R}^{L_v \times H_v}$ and text feature

$f_l \in \mathbb{R}^{L_l \times H_l}$ , respectively, where $H, W$ are the image size, $H_v$ and $H_l$ are the visual and text hidden embedding dimension, $L_v$ is the length of image token, which tokenized by a convolution projection, and $L_l$ is the length of text token, which tokenized by a lower-cased Byte Pair Encoding (BPE) with a 49,152 vocab size [52]. We extract the multi-level intermediate visual features $\{f_v^i\}_{i=1}^m \in \mathbb{R}^{m \times L_v \times H_v}$ and text features $\{f_l^i\}_{i=1}^n \in \mathbb{R}^{n \times L_l \times H_l}$, which obtained by the ViT block and text Transformer block, respectively, where $m$ and $n$ are the numbers of extracted layers.

Simultaneously, to reduce the inconsistency between the visual features of the uni-modal image backbone and those required for grounding, we introduce a multi-layer adaptive cross-modal bridge to the visual encoder that bridges image and text modalities. Each layer of the bridge has a learnable sample-agnostic weighting module, thus enabling the uni-modal visual backbone to perceive hierarchical cross-modal text features.

Additionally, to prevent the accumulation and amplification of perceptual errors in the visual encoder, we propose a hierarchical low-rank adaptation (HiLoRA) paradigm to adapting the pre-trained frozen parameters. During HiLoRA training, the entire adaptation process learns from shallow to deep layers. The gradients backward from the grounding encoder are updated hierarchically and adaptively into the low-rank matrix based on both visual features and hierarchical language features. Besides, the intermediate visual features are aggregated and fed to the grounding encoder, which not only benefits the perception of multi-level visual features but also facilitates direct gradient backward updates without going from deep to shallow in the HiLoRA low-stage training.

Finally, in the grounding encoder, we concatenate the multi-level visual features along with the hidden dimension, and leverage the weight $W_{mvp} \in \mathbb{R}^{(m \cdot H_v) \times H_g}$ of a MLP-based visual perceiver to project them into embedding space $g_v \in \mathbb{R}^{L_v \times H_g}$ with dimension $H_g$ to perceive multi-level visual representations:

$$g_v = \text{concat}[f_v^1, f_v^2, \cdots, f_v^m] \otimes W_{mvp}. \quad (1)$$

To prevent any perturbation on $[EOS]$ token and ensure the subsequent constraints remain unaffected, we exclusively utilize the linear projection features $g_l \in \mathbb{R}^{L_l \times H_g}$ of the last layer's text features $f_l^{last}$ to feed the grounding encoder. Finally, the input tokens of the grounding encoder are as follows:

$$x_g = [g_r, cls, \underbrace{g_v^1, g_v^2, g_v^3, \cdots, g_v^{L_v}}_{\text{CLIP image tokens } g_v}, \underbrace{g_l^1, g_l^2, g_l^3, \cdots, g_l^{L_l}}_{\text{CLIP text tokens } g_l}], \quad (2)$$

where $cls$ represents the classification token $[CLS]$, $g_r$ represents the learnable $[REG]$ token, which is used to output the regression results [9]. The $[EOS]$ token is the end token of each sequence within $f_l$ and $g_l$. The regression head is employed to conduct bounding box regression, which is a three-layer MLPs [9], each consisting of a linear layer and a ReLU activation layer. It outputs the final coordinate of the predicted grounding box $\hat{\mathcal{B}} = (\hat{x}, \hat{y}, \hat{w}, \hat{h})$.

## 3.2 Multi-layer Adaptive Cross-modal Bridge

The visual encoder of CLIP independently encodes the image, and the obtained multi-level visual features may be inconsistent with those required for grounding. Additionally, as inspired by [3, 8], specific layers of a pre-trained model may exhibit distinct responses to certain concepts or semantics that are independent of the input

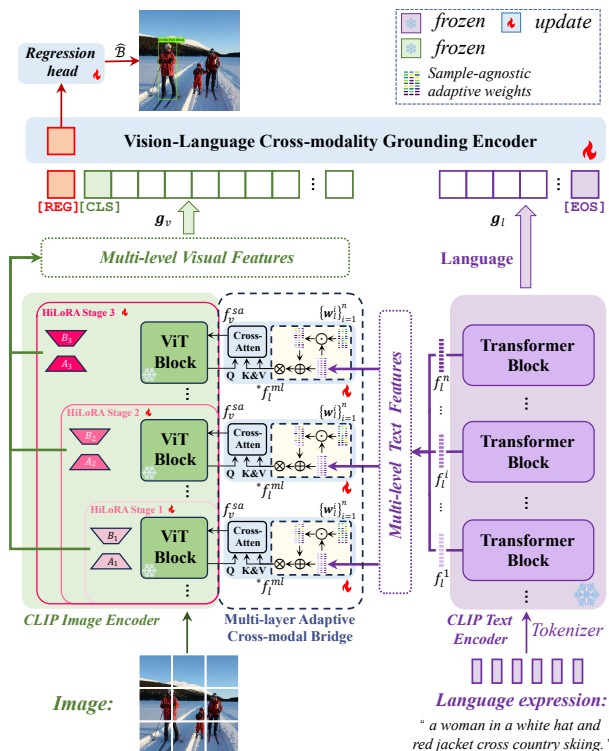

**Figure 2: Schematic representation of the hierarchical multimodal fine-grained modulation framework.**

and relevant to the network layers. Therefore, we should provide a wide range of multi-level text features for different visual layers to select and calibrate. Thus, to address these issues, we propose integrating a multi-layer adaptive cross-modal bridge into the image encoder to achieve fine-grained visual features.

The multi-layer adaptive cross-modal bridge (MACB) mainly consists of a sample-agnostic semantic weighting module and a multi-head cross-attention. It is inserted into specific ViT blocks, and we define the layer index set $C$ as the insertion positions. The sample-agnostic weighting module enables distinct hierarchical language feature perception among different layers. Specifically, we first extract and aggregate the intermediate language features $\{f_l^i\}_{i=1}^n \in \mathbb{R}^{n \times L_l \times H_l}$. Then, to stably strengthen or weaken the text features preferred by different visual layers, we utilize a residual operation to achieve selection of multi-level text features:

$$^*f_l^i = w_l^i \odot f_l^i + f_l^i. \quad (3)$$

where $\{w_l^i\}_{i=1}^n \in \mathbb{R}^{n \times L_l \times H_l}$ represent the learnable multi-level sample-agnostic adaptive weights within different layers, which can promote different visual layers respond distinctly to specific textual concepts or semantics. The weighted features are obtained by dot product between the sample-agnostic adaptive weights and multi-level features. We then add the weighted features to the original features to obtain the calibrated text features $\{^*f_l^i\}_{i=1}^n$. Subsequently, we concatenate and project them into visual embedding space $^*f_l^{ml} \in \mathbb{R}^{L_l \times H_v}$ to perceive multi-level language representations with linear projection weight $W_{proj} \in \mathbb{R}^{(n \cdot H_l) \times H_v}$:

$$^*f_l^{ml} = \text{concat}[^*f_l^1, ^*f_l^2, \cdots, ^*f_l^n] \otimes W_{proj}. \quad (4)$$

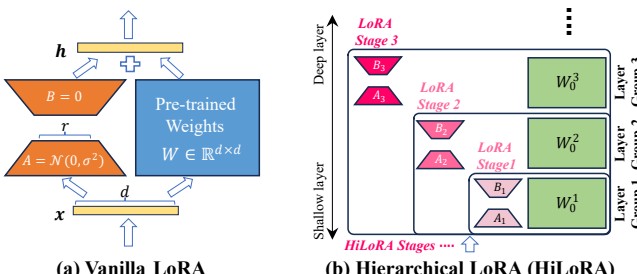

**(a) Vanilla LoRA**  **(b) Hierarchical LoRA (HiLoRA)**

**Figure 3: HiLoRA and vanilla LoRA. (a) The vanilla LoRA learns the global low-rank matrix utilizing the entire set of pre-trained weights in a single round. (b) The proposed HiLoRA employs a hierarchical approach to adapt the pre-trained model in a progressive manner, thereby finely reducing the task gap between pre-training and transfer tasks.**

Finally, we perform a multi-head cross-attention on the calibrated multi-level text features $^*f_l^{ml}$ (as key and value) and the layer-normalized visual features outputted by the self-attention in the ViT block (as query). Then, we add the resulting semantic-aware visual features $f_v^{sa}$ back to the block as residuals after a FFN operation.

## 3.3 Hierarchical Low-Rank Adaptation

Although the cross-modal bridge enables the visual encoder to incorporate language information, its residual connection manner cannot adapt the frozen parameters of the pre-trained model. As a result, there is still a discrepancy between the visual features and those required for grounding, which may lead to cumulative and amplified perceptual errors layer by layer. LoRA [18] presents a potentially feasible solution. However, as clarified in the Sec. 1, the vanilla LoRA performs one-round learning also cannot address these issues. To avoid cumulative and amplified perceptual errors, we need to design a hierarchical adaptation paradigm.

Instead of directly training specific dense layers in a neural network, vanilla LoRA [18] indirectly optimizes the rank-decomposition matrices of the changes occurring in dense layers while keeping the pre-trained weights frozen. As depicted in Fig. 3-(a), based on the vanilla LoRA definition, we can substitute the weight updates for a pre-trained weight $W_0 \in \mathbb{R}^{d \times k}$ with a low-rank decomposition $W_0 + \Delta W = W_0 + BA$, where $B \in \mathbb{R}^{d \times r}, A \in \mathbb{R}^{r \times k}$, and $r \ll \min(d, k)$, *i.e.*, the low rank $r$ is much smaller than the dimension $(d, k)$ of the original model. Throughout training, $W_0$ remains frozen, while $A$ and $B$ encompass trainable parameters. For hidden state $h = W_0 x$, the forward procedure can be formulated as:

$$h = W_0 x + \Delta W x = W_0 x + BAx. \tag{5}$$

To consider the hierarchical scenario, we first define two concepts, *i.e.*, layer group and LoRA stage. **Layer group** represents the divisions of the pre-trained network layers, while **LoRA stage** represents the execution of a small LoRA operation. By dividing the network layers of the pre-trained model into multiple layer groups, the learning of LoRA is divided into multiple stages where each stage relates to several layer groups. As depicted in Fig. 3-(b), from a hierarchical perspective, Hierarchical LoRA (HiLoRA) structure enables downstream task adaptation progressively from the shallow to deep layer within the network with multiple LoRA stages.

Specifically, we define the total layers of the pre-trained network as $L$, and then divide it into $G$ groups, each containing $L/G$ layers. Then, we denote $W_0^l \in \mathbb{R}^{d \times k}$ as the pre-trained weights of $l^{th}$ layer block in the network, where $l \in [1, L]$. We utilize LoRA $j$ ($1 \le j \le G$) to represent the $j^{th}$ adaptation stage. We denote the low-rank matrices of HiLoRA at the $j^{th}$ stage of the $l^{th}$ layer block as $A_j^l$ and $B_j^l$, and $A_j$ contains $\{A_j^l\}_{l=1}^{j \cdot L/G}$, $B_j$ contains $\{B_j^l\}_{l=1}^{j \cdot L/G}$, then each LoRA stage $j$ will update the low-rank matrices of $A_j$ and $B_j$. We denote $h_j^l$ as the hidden state $h$ at HiLoRA $j^{th}$ stage of the $l^{th}$ layer block. Then, the forward process of HiLoRA in each hidden state $h_j^l$ ($j \in [1, G]$) can be formulated as:

$$h_j^l = \begin{cases} W_0^l x^l, & when\ l > j \cdot L/G, \\ W_0^l x^l + \sum_{k=\lceil l \cdot G/L \rceil}^{j} B_k^l A_k^l x^l, & when\ l \le j \cdot L/G, \end{cases} \tag{6}$$

where $\lceil \cdot \rceil$ indicates rounding up to an integer, and $\lceil l \cdot G/L \rceil$ stands for calculating the index of layer groups in which $l^{th}$ layer is located.

With the assistance of the hierarchical mechanism, we can achieve better multimodal low-rank adaptation of multi-level visual features by utilizing textual semantic-aware visual features provided by the adaptive cross-modal bridge. Specifically, the layer groups of HiLoRA are associated with the insertion positions $C$ of the bridge. When $l > j \cdot L/G$, the forward process of HiLoRA in each hidden state $h_j^l$ can be formulated as:

$$h_j^l = \begin{cases} W_0^l f_v^{l-1}, & when\ l \notin C, \\ W_0^l (f_v^{l-1} + f_v^{sa}), & when\ l \in C. \end{cases} \tag{7}$$

While in $l \le j \cdot L/G$, the process can be formulated as:

$$h_j^l = \begin{cases} W_0^l f_v^{l-1} + \sum_{k=\lceil l \cdot G/L \rceil}^{j} B_k^l A_k^l f_v^{l-1}, & when\ l \notin C, \\ W_0^l (f_v^{l-1} + f_v^{sa}) + \sum_{k=\lceil l \cdot G/L \rceil}^{j} B_k^l A_k^l (f_v^{l-1} + f_v^{sa}), & when\ l \in C. \end{cases}$$
$$\tag{8}$$

During the backward process, the updates are gradually performed from $1^{st}$ to $G^{th}$ stage, and the learning rate can vary at different stages. Additionally, we use a random Gaussian initialization for $A$ and 0 for $B$, so $\Delta W = BA$ is 0 at the beginning of training. We then scale $\Delta W x$ by $\frac{\alpha}{r}$, where $\alpha$ is a constant in $r$. To mitigate inference latency or parameter increase, we incorporate the low-rank matrix into the pre-trained weights after every training stage.

HiLoRA provides a new interaction for refining latent representation, preventing direct gradient propagation of vanilla LoRA from deep to shallow layers. Simultaneously, through its hierarchical mechanism, it can avoid the accumulation of perceptual errors in the fine-tuning process, enabling fine-grained cross-modal interaction. Finally, it is worth noting that HiLoRA represents a basic hierarchical adaptation paradigm that is task-agnostic.

## 3.4 Training Objectives

To ensure the features learned by the cross-modal hierarchical structure meet the fine-grained and regional properties, we design multiple constraints to facilitate the training of HiVG framework. **Contrastive Learning Constraint.** To enhance training stability, we employ image-text Contrastive Learning (CL) as a constraint for HiLoRA. CL can also be formed between the grounding expression and the images within a shuffled training batch when differences are adequate. We treat the grounding image-text pairs as positive and all other random pairs as negative. We minimize the sum of two losses, one for text-to-image matching:

**Table 1: Comparison with latest SOTA methods on RefCOCO/+/g [41, 75], ReferItGame [21] and Flickr30k Entities [46] for grounding task. \* represents utilizing ImageNet [25] pre-training. † indicates that all of the RefCOCO/+/g training data has been used during pre-training. RN101, DN53, Swin-S, and ViT-B are shorthand for the ResNet101, DarkNet53, Swin-Transformer Small, and ViT Base, respectively. The latest CLIP-based SOTA methods are shaded in** gray **. We highlight the best performance of the base model in the** red **colors and bold the best results for the large model.**

| Methods | Venue | Visual Backbone | Language Backbone | Multi-task | RefCOCO val | testA | testB | RefCOCO+ val | testA | testB | RefCOCOg val | test | ReferIt test | Flickr test |
|---|---|---|---|---|---|---|---|---|---|---|---|---|---|---|
| Fine-tuning w. uni-modal pre-trained close-set detector and language model: (traditional setting) | | | | | | | | | | | | | | |
| TransVG [9] | ICCV'21 | RN101+DETR | BERT-B | ✗ | 81.02 | 82.72 | 78.35 | 64.82 | 70.70 | 56.94 | 68.67 | 67.73 | 70.73 | 79.10 |
| SeqTR [78] | ECCV'22 | DN53 | BiGRU | ✗ | 81.23 | 85.00 | 76.08 | 68.82 | 75.37 | 58.78 | 71.35 | 71.58 | 69.66 | 81.23 |
| RefTR* [27] | NeurIPS'21 | RN101+DETR | BERT-B | ✓ | 82.23 | 85.59 | 76.57 | 71.58 | 75.96 | 62.16 | 69.41 | 69.40 | 71.42 | 78.66 |
| Word2Pix [76] | TNNLS'22 | RN101+DETR | BERT-B | ✗ | 81.20 | 84.39 | 78.12 | 69.74 | 76.11 | 61.24 | 70.81 | 71.34 | – | – |
| QRNet [71] | CVPR'22 | Swin-S[39] | BERT-B | ✗ | 84.01 | 85.85 | 82.34 | 72.94 | 76.17 | 63.81 | 71.89 | 73.03 | 74.61 | 81.95 |
| VG-LAW [55] | CVPR'23 | ViT-Det [28] | BERT-B | ✗ | 86.06 | 88.56 | 82.87 | 75.74 | 80.32 | 66.69 | 75.31 | 75.95 | 76.60 | – |
| TransVG++[10] | TPAMI'23 | ViT-Det [28] | BERT-B | ✗ | 86.28 | 88.37 | 80.97 | 75.39 | 80.45 | 66.28 | 76.18 | 76.30 | 74.70 | 81.49 |
| Fine-tuning w. vision-language self-supervised pre-trained model: | | | | | | | | | | | | | | |
| CLIP-VG [63] | TMM'23 | CLIP-B | CLIP-B | ✗ | 84.29 | 87.76 | 78.43 | 69.55 | 77.33 | 57.62 | 73.18 | 72.54 | 70.89 | 81.99 |
| JMRI [79] | TIM'23 | CLIP-B | CLIP-B | ✗ | 82.97 | 87.30 | 74.62 | 71.17 | 79.82 | 57.01 | 71.96 | 72.04 | 68.23 | 79.90 |
| Dynamic-MDETR | TPAMI'23 | CLIP-B | CLIP-B | ✗ | 85.97 | 88.82 | 80.12 | 74.83 | 81.70 | 63.44 | 74.14 | 74.49 | 70.37 | 81.89 |
| **HiVG (ours)** | ACM MM'24 | CLIP-B | CLIP-B | ✗ | 87.32 | 89.86 | 83.27 | 78.06 | 83.81 | 68.11 | 78.29 | 78.79 | 75.22 | 82.11 |
| **HiVG-L (ours)** | ACM MM'24 | CLIP-L | CLIP-L | ✗ | **88.14** | **91.09** | **83.71** | **80.10** | **86.77** | **70.53** | **80.78** | **80.25** | **76.23** | **82.16** |
| Fine-tuning w. box-level dataset-mixed open-set detection pre-trained model / multi-task mix-supervised pre-trained model: | | | | | | | | | | | | | | |
| MDETR † [20] | ICCV'21 | RN101+DETR | RoBERT-B | ✗ | 86.75 | 89.58 | 81.41 | 79.52 | 84.09 | 70.62 | 81.64 | 80.89 | – | 83.80 |
| YORO† [16] | ECCV'22 | ViLT [24] | BERT-B | ✗ | 82.90 | 85.60 | 77.40 | 73.50 | 78.60 | 64.90 | 73.40 | 74.30 | 71.90 | – |
| DQ-DETR † [32] | AAAI'23 | RN101+DETR | BERT-B | ✗ | 88.63 | 91.04 | 83.51 | 81.66 | 86.15 | 73.21 | 82.76 | 83.44 | – | – |
| Grounding-DINO† | Arxiv'23 | Swin-T | BERT-B | ✗ | 89.19 | 91.86 | 85.99 | 81.09 | 87.40 | 74.71 | 84.15 | 84.94 | – | – |
| UniTAB † [69] | ECCV'22 | RN101+DETR | RoBERT-B | ✓ | 86.32 | 88.84 | 80.61 | 78.70 | 83.22 | 69.48 | 79.96 | 79.97 | – | 79.38 |
| OFA-B † [60] | ICML'22 | OFA-B | OFA-B | ✓ | 88.48 | 90.67 | 83.30 | 81.39 | 87.15 | 74.29 | 82.29 | 82.31 | – | – |
| OFA-L † [60] | ICML'22 | OFA-L | OFA-L | ✓ | 90.05 | 92.93 | 85.26 | 85.80 | 89.87 | **79.22** | 85.89 | 86.55 | – | – |
| **HiVG† (ours)** | ACM MM'24 | CLIP-B | CLIP-B | ✗ | 90.56 | 92.55 | 87.23 | 83.08 | 89.21 | 76.68 | 84.52 | 85.62 | 77.75 | 82.08 |
| **HiVG-L† (ours)** | ACM MM'24 | CLIP-L | CLIP-L | ✗ | **90.77** | **92.94** | **88.03** | **86.78** | **89.91** | 78.02 | **86.61** | **86.60** | **78.16** | **82.63** |

$$\mathcal{L}_{t2i} = -\frac{1}{N}\sum_{i}^{N}\log\frac{\exp(<\boldsymbol{t}_i^\top, \boldsymbol{v}_i>/\tau)}{\sum_{j=1}^{N}\exp(<\boldsymbol{t}_i^\top, \boldsymbol{v}_j>/\tau)}, \quad (9)$$

and the other for image-to-text matching:

$$\mathcal{L}_{i2t} = -\frac{1}{N}\sum_{i}^{N}\log\frac{\exp(<\boldsymbol{v}_i^\top, \boldsymbol{t}_i>/\tau)}{\sum_{j=1}^{N}\exp(<\boldsymbol{v}_i^\top, \boldsymbol{t}_j>/\tau)}, \quad (10)$$

where $N$ is the batch size, $\boldsymbol{v}_i$ and $\boldsymbol{t}_j$ are the normalized embeddings of image in $i^{th}$ pair and that of text in $j^{th}$ pair, respectively. $\tau$ is the temperature to scale the logits, and $< \cdot, \cdot >$ denotes cosine similarity operation. Therefore, the constraint can be formulated as:

$$\mathcal{L}_{CLC} = (\mathcal{L}_{t2i} + \mathcal{L}_{i2t})/2. \quad (11)$$

**Region-Text Contrastive Constraint.** Inspired by the image-level contrastive learning, we attempt to construct token-wise region-text contrastive constraint using ground truth bounding box as a mask to simulate text-to-image matching. Specifically, we extract text aggregation features, i.e., the $[EOS]$ token $\boldsymbol{t}_{eos}$, from grounding encoder and compute the similarity $\boldsymbol{s}_i$ with each visual token $\boldsymbol{v}_i$ after applying normalization and an MLP projection:

$$\boldsymbol{s}_i = \sigma(<\boldsymbol{t}_{eos}^\top, \text{MLP}(\boldsymbol{v}_i)>), \; i = 1, 2, ..., L_v, \quad (12)$$

where $\sigma$ denotes the sigmoid function. Tokens within the bounding box are considered as positive, while those outside are regarded as negative. Subsequently, we employed Focal loss [30] and Dice/F-1 loss [43] to constrain the aggregated similarity $\boldsymbol{s} = (\boldsymbol{s}_1, \boldsymbol{s}_2, ..., \boldsymbol{s}_{L_v})$ and the nearest downsampling box mask $\boldsymbol{m}_d \in \mathbb{R}^{1 \times H/P \times W/P}$:

$$\mathcal{L}_{RTCC} = \lambda_{focal}\mathcal{L}_{focal}(\boldsymbol{s}, \boldsymbol{m}_d) + \lambda_{dice}\mathcal{L}_{dice}(\boldsymbol{s}, \boldsymbol{m}_d), \quad (13)$$

where $\lambda_{focal}$ and $\lambda_{dice}$ are the coefficients to control the two loss functions, and $P$ is the patch size.

**Training Loss.** The box regression loss is formulated by leveraging smooth L1 loss [13] and Giou loss [50] with coefficient $\lambda_{l_1}$ and $\lambda_{giou}$:

$$\mathcal{L}_{BOX} = \lambda_{l_1}\mathcal{L}_{\text{smooth-l1}}(\hat{\mathcal{B}}, \mathcal{B}) + \lambda_{giou}\mathcal{L}_{\text{giou}}(\hat{\mathcal{B}}, \mathcal{B}), \quad (14)$$

where $\mathcal{B}$ donates the ground truth box. Finally, the overall training loss of the model is determined by the sum of the regression loss and the two framework constraints:

$$\mathcal{L}_{total} = \mathcal{L}_{BOX} + \mathcal{L}_{CLC} + \mathcal{L}_{RTCC}. \quad (15)$$

## 4 EXPERIMENTS

### 4.1 Implementation Details

**Datasets and Evaluation Metrics.** The effectiveness of our method is validated on five widely utilized datasets, namely the three REC datasets (RefCOCO/+/g [41, 75]), as well as two PG datasets (Refer-ItGame [21] and Flickr30k Entities [46]). In PG, the query pertains to a specific phrase, while in REC, the query refers to a referring expression. The text of RefCOCO+/g exhibits greater length and complexity in comparison to that of RefCOCO. We follow the previous researches that employs Intersection-over-Union (IoU) as the evaluation metric. Specifically, a prediction is deemed accurate only when its IoU exceeds or equals 0.5. Finally, we compute the prediction accuracy for each dataset as a performance indicator.

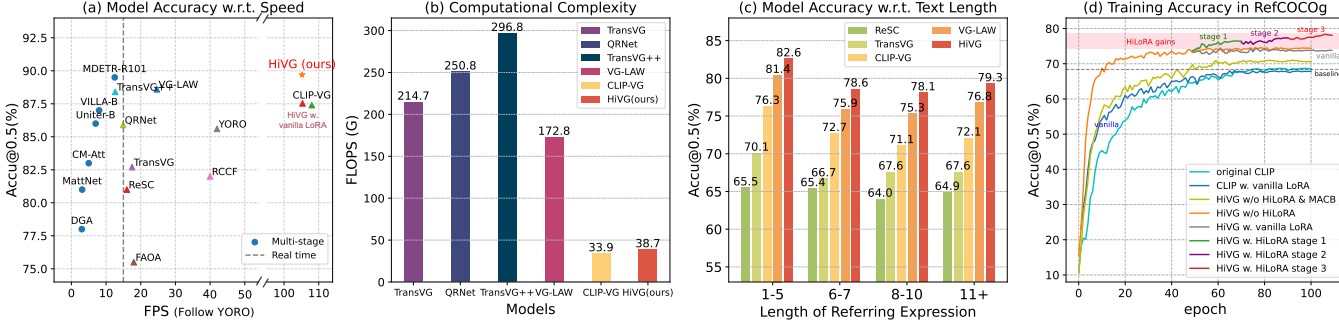

Figure 4: Comparison between HiVG (base) and SOTA models, as well as the ablation study of HiVG on the main modules. (a) HiVG achieves significant energy efficiency advantages, 8.2× faster than TransVG++[10] while outperforming it on RefCOCO-val. (b) The computational complexity of HiVG is only 13.0% compared with TransVG++. (c) HiVG outperforms SOTA models in different expression lengths on RefCOCOg-test. (d) HiLoRA method brings significant performance gains to HiVG model.

Table 2: Training/inference cost comparison. The results are obtained on RefCOCO dataset. † indicates that the model's code is not publicly available, and the replicated estimation results are shown. (*FPS: images / (GPU · second)*)

| Model | update/all param. | update ratio | Flops (G)↓ | train FPS↑ | test FPS↑ | testA time↓ | testA Acc.↑ |
|---|---|---|---|---|---|---|---|
| TransVG | 168/170M | 98.8% | 214.7 | 22.85 | 59.55 | 95 s | 82.7 |
| QRNet | 273/273M | 100% | 250.8 | 9.41 | 50.96 | 111 s | 85.9 |
| VG-LAW† | 150/150M | 100% | 172.8 | – | 83.9 | – | 88.6 |
| CLIP-VG | 21/181M | 12.2% | 33.9 | 252.6 | 377.8 | 15 s | 87.8 |
| TransVG++† | 171/171M | 100% | 296.8 | – | 43.1 | – | 88.4 |
| **HiVG**(ours) | 41/206M | 20.1% | 38.7 | 239.6 | 354.6 | 16 s | 89.9 |

**Network Architecture.** We employed CLIP ViT-B/16 and CLIP ViT-L/14 as the backbone of our HiVG-B (default) and HiVG-L versions. In the base version, the HiLoRA module utilizes a rank of 32 and an $\alpha$ coefficient of 16. The encoder layers are evenly divided into 3 groups, and HiLoRA is applied with 3 stages accordingly. HiVG extracted $1^{th}$, $4^{th}$, $8^{th}$, and $12^{th}$ layer features of the visual encoder, the cross-modal bridge injected $4^{th}$, $8^{th}$, and $12^{th}$ layer, and text aggregated from $1^{th}$ to $12^{th}$ layer features of the text encoder. In the grounding encoder, we adopted the pre-norm instead of the post-norm structure and set the hidden dimensions as the same with text encoder.

**Training Details.** To prevent catastrophic forgetting, we freeze the original parameters of CLIP's two encoders. Since the parameters of the low-stage HiLoRA are included in the high-stage HiLoRA, our updated parameters do not show any increase compared to the vanilla LoRA. Besides, HiLoRA represents a PEFT approach for the pre-trained model, and the grounding encoder employs random Xavier initialization. Thus, to enhance training stability, we perform training in two stages. In the first stage, we trained the grounding encoder, regression head at a high learning rate without activating HiLoRA. It is imperative to employ HiLoRA for the text encoder with only one layer group as well, in order to mitigate the risk of catastrophic forgetting. The batch size is set to 60. Our model is optimized end-to-end by using the AdamW optimizer and a cosine learning scheduler with an initial learning rate of $2.5 \times 10^{-4}$ for 50 epochs during the first stage. During HiLoRA adaptation, the learning rates in three stages are $1.0 \times 10^{-4}$, $0.5 \times 10^{-4}$, and $0.25 \times 10^{-4}$ with 20 epochs, respectively. Besides, to ensure a fair

comparison, like the existing works [10, 55], we pre-perform a vanilla LoRA adapting of CLIP's image encoder under ViT-Det [28] detection framework on MSCOCO dataset, with excluding the validation and test images of RefCOCO/+/g. Our framework and experiments are based on PyTorch by using 8 NVIDIA A100 GPUs.

## 4.2 Comparison with State-of-the-Art Methods

**Experimental Setting.** It is worth emphasizing that, as described in Sec. 2.1, our focus is on the transfer learning of self-supervised pre-trained models for grounding tasks. *(1)* We follow the basic fine-tuning setting with the same as CLIP-VG [63] and Dynamic-MDETR [53], *etc.*. *(2)* In particular, we also compare with the traditional setting of fine-tuning with pre-trained detection models (*e.g.*, TransVG [9], TransVG++[10], *etc.*). *(3)* Additionally, we also follow the previous works that utilized a dataset-mixed pre-training setting (*e.g.*, MDETR [20], OFA[60]) and mix the training data (only includes the RefCOCO/+/g, ReferIt, Flickr30k datasets) for intermediate pre-training. This allows us to compare our results with these works in a relatively fair manner. The details are presented in Tab. 1.

**RefCOCO/RefCOCO+/RefCOCOg/ReferIt/Flickr.** As presented in Tab. 1, we compare our results on five widely used datasets with the latest SOTA works, including CLIP-VG [63], Dynamic-MDETR [53], TransVG++[10], grounding-DINO [33] and OFA [60] *etc.*. *(1)* **When compared to the CLIP-based fine-tuning SOTA work**, *i.e.*, Dynamic-MDETR, our approach consistently outperforms it by achieving an increase of 3.15%(testB), 2.11%(testA), 4.30%(test), 4.85%(test), 0.22%(test) on all five datasets. *(2)* **When compared to the detector-based fine-tuning SOTA work**, *i.e.*, TransVG++, our approach demonstrates superior performance (improved by 2.30%(testB), 3.36%(testA), 2.49%(test), 0.52%(test), 0.62%(test)) across all five datasets. The improvement of our results on the RefCOCO+/g datasets is considerably more significant, indicating our model exhibits a stronger capacity for semantic comprehension in complex sentences. *(3)* **When compared with the dataset-mixed pre-training works**, the base model of our work outperforms Grounding-DINO [33] by 1.24%(testB), 1.81%(testA), and 0.68%(test) on the RefCOCO/+/g datasets, and it also outperforms OFA [60] by 3.93%(testB), 2.06%(testA), and 3.31%(test). After dataset-mixed pre-training, our performance has significantly improved, further demonstrating the effectiveness of our method.

**Table 3: Ablation study of the main modules, includes Multi-layer Adaptive Cross-modal Bridge (MACB) and HiLoRA.**

| MACB | HiLoRA | Accu@0.5(%) val | test |
|------|--------|------|------|
| ✗ | ✗ | 73.48 | 73.01 |
| ✓ | | 76.53 | 75.77 |
| | ✓ | 76.41 | 76.12 |
| ✓ | ✓ | **78.29** | **78.79** |

**Table 4: Ablation study on the implementation of multi-layer adaptive cross-modal bridge (MACB) on RefCOCOg dataset. *w/o* denotes without, and *w.* denotes with. (Accu@0.5(%))**

| Architecture | val | test |
|--------------|-----|------|
| MACB *w/o.* sample-agnostic weights | 75.43 | 74.87 |
| MACB *w/o.* cross-attention module | 74.29 | 74.18 |
| MACB *w.* weights' shape $1 \times 1 \times H_l$ | 74.81 | 74.38 |
| MACB *w.* weights' shape $n \times 1 \times H_l$ | 77.08 | 77.42 |
| MACB *w.* weights' shape $n \times L_l \times 1$ | 77.42 | 78.49 |
| **MACB *w.* weights' shape $n \times L_l \times H_l$** | **78.29** | **78.79** |
| MACB *w.* layer-to-layer linear connect | 76.51 | 76.30 |
| MACB *w.* only last layer of text features | 77.07 | 76.82 |

**Table 5: Ablation study of different components in HiLoRA on RefCOCOg-test. $r$ represents the value of low rank.**

| Architecture | Accu@0.5(%) |
|--------------|-------------|
| HiLoRA three-stage-$1^{th}$ ($r$=32) | 76.39 |
| HiLoRA three-stage-$2^{th}$ ($r$=32) | 77.87 |
| **HiLoRA three-stage-$3^{th}$ ($r$=32)** | **78.79** |
| HiLoRA two-stage ($r$=32) | 77.97 |
| HiLoRA four-stage ($r$=32) | 78.16 |
| HiLoRA three-stage ($r$=16) | 77.57 |
| HiLoRA three-stage ($r$=64) | 76.90 |
| HiLoRA deep-to-shallow layer | 73.93 |

**Parameter, Training/Inference Costs and Efficiency.** As shown in Tab. 2, Fig. 4-(a) and (b), HiVG achieves significant energy efficiency advantages, **8.2×** faster than TransVG++ while outperforming it on RefCOCO. The computational complexity of HiVG model is **only 13.0**% compared with TransVG++.

**Analysis of Referring Expression Length.** As shown in Fig. 4-(c), we conducted a comparison of different expression lengths on the RefCOCOg dataset. It shows that HiVG exhibits superior comprehension for longer and more complex texts, while its performance remains stable as text length increases. Furthermore, compared to CLIP-VG, our method demonstrates significantly better results.

### 4.3 Ablation Study

**Ablation Study of the Main Modules.** We conducted the ablation study on RefCOCOg datasets. As presented in Tab. 3 and Fig. 4-(d), our MACB and HiLoRA modules enhances performance by 3.05% and 2.93%. Our hierarchical adaptation structure facilitates fine-grained alignment and interaction between visual and textual modal features, significantly boosting the grounding performance.

**Ablation Study of MACB.** As shown in Tab. 4, we conducted an ablation study on the implementation of the multi-layer adaptive

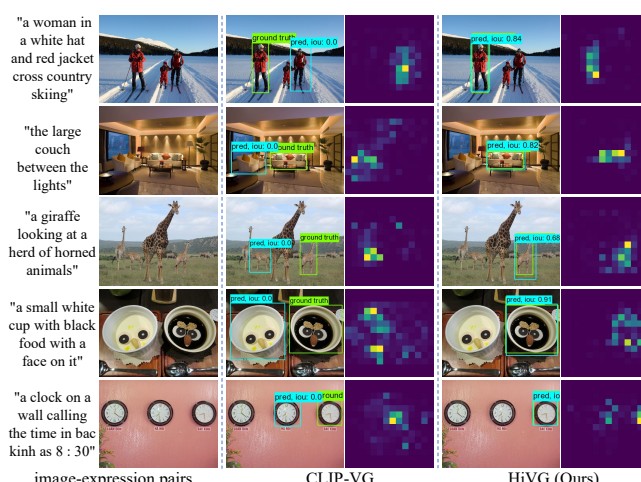

image-expression pairs     CLIP-VG     HiVG (Ours)

**Figure 5: Qualitative results of our HiVG and CLIP-VG models on RefCOCOg-val datasets. We present the prediction box with IoU (in cyan) and the ground truth box (in green) in a unified image to visually display the grounding accuracy.**

cross-modal bridge (MACB, default using 12 layers of text features). The weights in the table denotes the sample-agnostic weights. The table shows that our designed structure can effectively utilize multi-level text features and achieve hierarchical adaptation.

**Ablation Study of HiLoRA.** As presented in Tab. 5 and Fig. 4-(d), we conducted an ablation study on HiLoRA with different LoRA stages and various low ranks. It is observed that employing 3-stage HiLoRA with low rank as 32 achieves the best performance.

### 4.4 Qualitative Results

We visually present the results of several relatively challenging examples in Fig. 5. The attentions show the [REG] token over vision tokens from the last grounding block of each model. HiVG demonstrates exceptional semantic understanding capabilities in the complex sentences.

## 5 CONCLUSION

In this paper, we introduce a hierarchical multimodal fine-grained modulation framework, namely HiVG, which effectively implements fine-grained adaptation of the pre-trained model in the complex grounding task. It is a concise and efficient end-to-end framework that can simultaneously alleviate two kinds of task gaps, *i.e.*, data bias and learning objectives, through a multi-layer adaptive cross-modal bridge and a hierarchical low-rank adaptation paradigm. Our exploration in hierarchical cross-modal features offer new insights for the future grounding research, which has been neglected in past works.

## ACKNOWLEDGMENTS

This work was supported in part by the National Natural Science Foundation of China under Grants 62036012, U23A20387, 62322212, 62072455, in part by Pengcheng Laboratory Research Project under Grant PCL2023A08, and also in part by National Science and Technology Major Project under Grant 2021ZD0112200.

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
