# OpenReview forum: "HiVG: Hierarchical Multimodal Fine-grained Modulation for Visual Grounding"
_acmmm.org/ACMMM/2024/Conference — MM2024 Poster_

### Official Review · Reviewer_SMYY · 2024-05-12

**Rating:** 5
**Confidence:** 4

**Summary:**

This paper studies the multimodal pre-training for visual grounding. To tackle the task gap between pre-training and grounding, this paper proposes a hierarchical multimodal fine-grained modulation framework. Concretely, a multi-layer adaptive cross-modal bridge and a hierarchical multimodal low-rank adaption paradigm are proposed to tackle the inconsistency between visual features and text-relevant regions,  and establish the connection between multi-level visual and text features, respectively. Extensive experiments conducted on the benchmark datasets exhibit the effectiveness of the proposed method.

**Strengths:**

- The proposed hierarchical multimodal low-rank adaption paradigm is innovative.
- This paper is well organized and easy to read.
- The proposed method exhibits superior performance in visual grounding.

**Limitations:**

This paper provides extensive experiments to validate the effectiveness of the proposed model. I am interested in the ablation study results of training loss in Table 4 in the supplementary material. The results indicate some instability in training when the CLC loss is omitted. Could the authors provide a more detailed discussion of the potential causes of this instability?

**Suitability:**

3

---

### Official Review · Reviewer_nq6h · 2024-05-23

**Rating:** 4
**Confidence:** 2

**Summary:**

This paper addresses data biases and differences in learning objectives within the visual grounding task, proposing a multi-layer adaptive cross-modal bridge and a hierarchical multimodal low-order adaptation (Hi LoRA) paradigm. This paradigm adjusts cross-modal features from shallow to deep layers, establishing connections between multi-level visual and textual features. Extensive experiments have demonstrated the effectiveness of HiVG.

**Strengths:**

1.The paper makes non-trivial advances over the current SOTA methods, the proposed Hi lora prevents  from shallow to deep layers.
2.This paper is well written, easy to follow.
3.The conducted experiments show that the designed HiVG outperforms related visual grounding methods.

**Limitations:**

1.I wonder whether HiVG is compatible with other multi-modal downstream tasks.
2.Some symbols mentioned in Section 3 do not appear in Figure 2, making it difficult to read.
3.The multi-layer adaptive cross-modal bridge facilitates interaction between textual and visual information. In the ablation studies provided by the author, only choices of related hyper-parameters are discussed, with no visual validations of how text aids in visual calibration.
4.In the experimental section, the author does not validate how Hi LoRA reduces perceptual errors.

**Suitability:**

3

---

### Official Review · Reviewer_Effd · 2024-05-24

**Rating:** 5
**Confidence:** 3

**Summary:**

In order to fully exploit the potential of a CLIP model for visual grounding tasks, this paper studies the problems of data bias and difference in learning objectives. To this end, the multi-level image and text representations are integrated from a hierarchical perspective with the facilitation of the developed multi-layer adaptive cross-modal bridges and the proposed hierarchical low-rank adaptation paradigm.

**Strengths:**

1.  The paper is well written and easy to follow. The diagrams are beautiful and helpful for understanding.
2.  Extensive experiments, including both quantitative and qualitative analysis, indicate the effectiveness of the proposed HiVG method. Additionally, HiVG also shows its superiority on model parameter size, training/inference costs and model efficiency.
3.  This paper provides a strong baseline for Clip-based visual grounding research.

**Limitations:**

It is a good paper, the design of Hi LoRA is interesting, I just have two small questions:

1.	The proposed cross-modal bridge module learns sample-agnostic adaptive weights within different layers, which have the same length of text token. So how do these weights work on the input texts with different lengths? Is it reasonable for using the same weights to reweight the different tokens as in Eq. (3)?

2.	The authors claim the motivation of designing the Hi LoRA module is “Since high-level features depend on low-level features, and they are susceptible to perturbations of the shallow features. If all layers of a large-scale pre-trained model are adapted simultaneously, perceptual errors in bottom layers may accumulate and amplify”. Although Hi LoRA achieves higher performance than vanilla LoRA, I am little confused that the underlying mechanism of Hi LoRA to solve the above issue. Why using such hierarchical structure can deal with this “error accumulation” problem?
Nevertheless, I generally like this paper, so I vote for accepting this paper.

**Suitability:**

3

---

### Official Review · Reviewer_ajV8 · 2024-05-25

**Rating:** 5
**Confidence:** 3

**Summary:**

This paper proposes a hierarchical fine-grained multimodal module network HiVG to bridge the data gap and training loss gap between multimodal pretraining and grounding tasks based on CLIP. HiVG includes multiple cross-modal bridges and hierarchical multimodal LoRA. The former can resolve the inconsistency between the visual features extracted by the pre-trained model and the visual features required for the grounding task, while establishing associations between multi-level visual and textual features. The latter prevents the accumulation of perceptual errors by adjusting cross-modal features from shallow to deep in a hierarchical manner. The experiments are conducted on five benchmark datasets, demonstrating the sota performance.

**Strengths:**

- The propsosed Hierarchical LoRA is new for progressivingly adapt the pre-trained model.
- The experimental results have achieved state-of-the-art (SOTA), comparied with the latest baselines.
- The ablation study is comprehensive.

**Limitations:**

- The placement of the "frozen" diagram in the light green CLIP Image Encoder part of Fig. 2 is somewhat confusing. I suggest that it should point to the dark green ViT Block part on the right.
- Hi LoRA can be connected as HiLoRa or Hi-LoRa for better representation.
- Several works are misssing, such as
[1] M. Lu, R. Li, F. Feng, Z. Ma and X. Wang, "LGR-NET: Language Guided Reasoning Network for Referring Expression Comprehension," in IEEE Transactions on Circuits and Systems for Video Technology, 2024, doi: 10.1109/TCSVT.2024.3374786.
[2] P. Miao, W. Su, G. Wang, X. Li and L. Xi, "Self-Paced Multi-Grained Cross-Modal Interaction Modeling for Referring Expression Comprehension," in IEEE Transactions on Image Processing, vol. 33, pp. 1497-1507, 2024, doi: 10.1109/TIP.2023.3334099.
[3] J. Ke, J. Wang, J. -C. Chen, I. -H. Jhuo, C. -W. Lin and Y. -Y. Lin, "CLIPREC: Graph-Based Domain Adaptive Network for Zero-Shot Referring Expression Comprehension," in IEEE Transactions on Multimedia, vol. 26, pp. 2480-2492, 2024, doi: 10.1109/TMM.2023.3297312.
- HiLoRa should be evaluted in other transfer tasks.
- The cross-modal birdge is not novel.

**Suitability:**

3

---

### Meta-Review · Area_Chair_y2Df · 2024-07-03

**Recommendation:** Accept (Poster)
**Confidence:** 5

**Metareview:**

This paper proposes a hierarchical multimodal fine-grained modulation framework, namely HiVG, for the challenging task of visual grounding.  HiVG consists of a multi-layer adaptive cross-modal bridge and a hierarchical multimodal low-rank adaptation paradigm. Extensive results on several popular datasets validate the effectiveness of the proposed method.

(+) On the positive side, the reviewers found the method to be interesting and effective and appreciate the good presentation and impressive results.

(-) On the negative side, the reviewers still have some concerns on the instability, insufficient discussion, missing related work, and clarity of the writing.

Several technical questions and suggestions were raised by the reviewers. The authors have taken these into consideration.
Overall, all reviewers agreed to accept the paper after the rebuttal. But most reviewers did not clearly and seriously justify their final ratings. Especially, most "Strengths" comments are short and simple. Besides, the "Confidence" of the Reviewer nq6h whose final rating is Accept is just "Somewhat Confident".

Therefore, the AC  leans towards recommending  Accept (Poster).